# Antibacterial Electrospun Polycaprolactone Nanofibers Reinforced by Halloysite Nanotubes for Tissue Engineering

**DOI:** 10.3390/polym14040746

**Published:** 2022-02-15

**Authors:** Viera Khunová, Mária Kováčová, Petra Olejniková, František Ondreáš, Zdenko Špitalský, Kajal Ghosal, Dušan Berkeš

**Affiliations:** 1Department of Plastics and Rubber, Faculty of Chemical and Food Technology, Slovak University of Technology, Radlinského 9, 81237 Bratislava, Slovakia; 2Polymer Institute, Slovak Academy of Sciences, Dúbravská Cesta 9, 84541 Bratislava, Slovakia; m.kovacova@savba.sk (M.K.); zdeno.spitalsky@savba.sk (Z.Š.); 3Department of Biochemistry and Microbiology, Slovak University of Technology, Radlinského 9, 81237 Bratislava, Slovakia; petra.olejnikova@stub.sk; 4Central European Institute of Technology, Brno University of Technology, Purkynova 656/123, 61200 Brno, Czech Republic; frantisek.ondreas@ceitec.vutbr.cz; 5CONTIPRO a.s., Dolní Dobrouč 401, 56102 Dolní Dobrouč, Czech Republic; 6Department of Pharmaceutical Technology, Jadavpur University, Kolkata 700032, India; kajal.ghosal@gmail.com; 7Department of Organic Chemistry, Slovak University of Technology, Radlinského 9, 81237 Bratislava, Slovakia; dusan.berkes@stuba.sk

**Keywords:** biocompatible, antibacterial, halloysite, erythromycin, polycaprolactone, nanofibers, electrospinning, tissue engineering

## Abstract

Due to its slow degradation rate, polycaprolactone (PCL) is frequently used in biomedical applications. This study deals with the development of antibacterial nanofibers based on PCL and halloysite nanotubes (HNTs). Thanks to a combination with HNTs, the prepared nanofibers can be used as low-cost nanocontainers for the encapsulation of a wide variety of substances, including drugs, enzymes, and DNA. In our work, HNTs were used as a nanocarrier for erythromycin (ERY) as a model antibacterial active compound with a wide range of antibacterial activity. Nanofibers based on PCL and HNT/ERY were prepared by electrospinning. The antibacterial activity was evaluated as a sterile zone of inhibition around the PCL nanofibers containing 7.0 wt.% HNT/ERY. The morphology was observed with SEM and TEM. The efficiency of HNT/ERY loading was evaluated with thermogravimetric analysis. It was found that the nanofibers exhibited outstanding antibacterial properties and inhibited both Gram- (*Escherichia coli*) and Gram+ (*Staphylococcus aureus*) bacteria. Moreover, a significant enhancement of mechanical properties was achieved. The potential uses of antibacterial, environmentally friendly, nontoxic, biodegradable PCL/HNT/ERY nanofiber materials are mainly in tissue engineering, wound healing, the prevention of bacterial infections, and other biomedical applications.

## 1. Introduction

Halloysite nanotubes (HNTs) are natural, nontoxic, biocompatible, eco-friendly, and low-cost materials recognized by the Environmental Protection Agency as nanomaterials (EPA 4). Presently, HNTs play a significant role in drug-carrier systems suitable for different biomedical applications, e.g., tissue engineering [1,2].

HNTs, naturally occurring in 1:1 layered aluminosilicate clay, consist of aluminum and silicon oxide layers rolled into tubes. The layers are rolled into tubes because of differences in the sizes of silicon and aluminum ions [3]. The typical length of the nanotubes is about 1–2 μm. Their outer and inner diameters range from 50 to 100 nm and from 10 to 50 nm, respectively [4,5]. For biomedical applications, the most significant advantages of HNTs compared to other tubular silicates are that they present a unique combination of structure, natural availability, rich functionality, good biocompatibility, and cytotoxicity [2,3,6,7,8,9].

One of the most remarkable features of HNTs is their different surface chemistries at the inner and outer sides of the tubes: silica sheets make up the external surfaces of the tubes and aluminum oxide makes up the inner (lumen) surface chemistry. Furthermore, alumina has a positive charge up to pH 8.5, and silica has a negative charge at pH values above 1.5 [10,11]. Due to differently charged outer and inner sides, it is possible to utilize HNTs as multifunctional nanocontainers for the selective modification of the outer and inner sides of nanotubes [7,11,12,13,14,15,16].

Halloysite’s inner diameter fits well to macromolecules and proteins [6]. In this regard, drugs of smaller molecular size are typically vacuum-loaded within the inner lumen of the nanotube, and drugs with larger molecular size can attach to the outer surface of the halloysite [17].

HNTs have also been successfully used as low-cost nanocontainers for several antibiotics such aminoglycoside gentamicin [18] and β-lactam antibiotic amoxicillin [19]. In addition, a wide range of applications of vancomycin-loaded halloysite nanotubes have been presented in alginate-based wound dressing [20] and silk fibroin hydrogel applicable for bone tissue engineering [21].

Electrospinning is a technology for the fabrication of continuous nanofibers with a simple setup [22]. Electrospun nanofibers can be prepared from natural or synthetic polymers or their blends. In the past decade, significant progress has been achieved in researching advanced electrospun nanofibers for biomedical applications or one-dimensional nanofibers made of intrinsically conducting polymers [23]. Recent advances in the electrospinning of functional scaffolds for tissue engineering and nanofiber scaffolds were summarized in the work of Hanumantharao et al. [24]. Simultaneously with the improvement of electrospun process technology, a lot of effort is being made to study new types of antimicrobial nanoparticles or make the known ones much more effective against microbial effects [25,26,27,28].

At present, there have been many examples where a biodegradable polymer matrix was combined with antimicrobial nanofillers (e.g., metals and/or metal oxides) for the preparation of polymer nanocomposites for biomedical applications [29,30,31]. An extensively used synthetic, biodegradable, semi-crystalline polymer used for biomedical applications is polycaprolactone (PCL) [32]. PCL is well known for its versatile use, biocompatibility, chemical stability, thermal stability, slow biodegradation (around 24 months), tissue compatibility, and easy processing [31,33]. PCL is often used in biomedicine as an FDA-approved material in the form of nanofibers that have evolved as controllable drug delivery systems [34,35]. Furthermore, due to its slow degradation rate, PCL is a preferred polymer mainly used as a long-term drug delivery carrier. Another advantage of PCL is its compatibility with a wide range of drugs, which provides a homogenous distribution of predominantly lipophilic drugs in the carrier matrix due to its hydrophobic nature [36]. Moreover, in our earlier work, it was confirmed that except for an important and strong reinforcing effect, all studied PCL/Gel nanofibers with an HNT content from 0.5 to 9.0 wt.% were non-toxic and had no effect on cell behavior [35]. Thus, through a combination of low-strength PCL with drug-loaded halloysite, it is possible prepare reinforced biodegradable polymer nanocomposites with regular drug release.

In this work, electrospun nanofibers based on PCL and drug-loaded HNTs were studied. HNTs were used as a nanocarrier for erythromycin (ERY) as a model antibacterial active compound with wide range of antibacterial activity on both Gram-positive and Gram-negative bacteria. ERY—a macrolide antibiotic used to treat a number of bacterial infections, e.g., respiratory tract infections and pelvic inflammatory disease—was used for HNT loading. In addition, erythromycin is clinically used in dermatology as a very effective topical antibiotic drug in treating bacterial skin disease.

ERY is soluble in methanol but has minimal solubility in an aqueous solution and presents acid instability, which limit its broader application. Therefore, drug-carrier systems based on HNTs and ERY have great potential to overcome these weaknesses. Moreover, cheap ERY was chosen as a model substrate for another type of pH-sensitive macrolide antibiotics intensively studied by our group [37].

## 2. Materials and Methods

### 2.1. Materials

PCL CAPA^®^ 6800 (M_w_ = 80,000) and Erythromycin E6376 were obtained from Sigma-Aldrich Saint Louis, MO 63103, USA and ULTRA HalloPure, respectively, produced by I-Minerals Inc., Vancouver, BC, Canada. The ULTRA HalloPure comprised purified HNTs with 93.5% of halloysite, 6.1% of kaolinite, and 0.4% of quartz from Dragon Mine in Utah of USA, produced by Applied Minerals Inc., New York, NY, USA. The dimensions are presented in Table 1. Methanol and distilled water were purchased from Lachner s.r.o., Neratovice, Czech Republic.

### 2.2. Loading of a Drug in Halloysite Nanotubes

The loading of active agents to halloysite is based on the diffusion of molecules from an external solution into the inner part of HNTs due to the concentration gradient. The evaporation of the solvent under the vacuum elevates the concentration of the active agents in the solution and enhances the diffusion rate. Therefore, fast-drying solvents with low viscosities such as acetone or ethanol are preferable for organic substances [38]. To eliminate potential water on the outer part of HNTs before the loading procedure, HNTs were dried in the oven for 2 h at 150 °C. Since ERY is soluble in methanol, after the drying, it was dissolved in methanol. HNTs were dispersed in the ERY solution and sonicated for 20 min. The rate of halloysite to erythromycin was 60:40. After the sonification, halloysite loaded with ERY was dried in a vacuum and washed with distilled water using a 0.2 µm membrane filter. Residual unloaded ERY was removed with methanol (4 h).

HNTs were dried in the oven for 2 h at 150 °C. We dissolved 400 mg of the drug in 25 mL of methanol. We dispersed 600 mg of dried halloysite in a drug solution (ratio 40:60 ERY:HNT), which was then sonicated for 20 min. Then, this dispersion was dried in a vacuum. After vacuum drying, it was washed with distilled water using a 0.2 µm membrane filter. Residual unloaded ERY was removed with methanol (4 h).

### 2.3. Optimization of Electrospinning Process Parameters

The electrospinning process for PCL composites (Figure 1) was optimized and previously described by our group [29]. Briefly, a solution of 10% w/v PCL was prepared in a mixture of chloroform and methanol in a ratio of 4:1 and continuously stirred for 2 h. Then, the mixture was kept unstirred for another 15 min to remove any bubbles present in the solution. Prepared HNTs loaded with ERY were added to the PCL solution and stirred for another 1 h. Electrospun fibers were fabricated using a Spellman high voltage power source (Spellman High Voltage Electronics Corporation, New York, NY, USA) and syringe pump (New Era Pump Systems, Inc., New York, NY, USA). The electrospinning apparatus consisted of a 10 mL syringe that was integrated with a grounded electrode, and the needle diameter was 0.41 mm. The distance between tee collector and source was kept to 13 cm. One thin aluminum sheet was fixed over the static collector. The feeding rate for the electrospinning solution was set to 1 mL/h at a voltage of 25 kV. The temperature for the location varied between 21 and 29 °C, and the humidity varied from 73% to 93%.

### 2.4. Scanning Electron Microscopy

For the needs of scanning electron microscopy (SEM), the sample’s surface was covered in gold with a Sample Preparation System Quorum Technologies Q150R S/E/ES sputter coater evaporator (Quorum Technologies, Laughton, England, and the micrographs were obtained with an FIB Microscope Quanta 3D 200i (FEI Company, Tokyo, Japan) in a secondary electron mode at different magnitudes. The surface morphologies of pure and modified fibers were obtained.

### 2.5. Transmission Electron Microscopy

The structure of HNTs loaded with ERY was characterized by transmission electron microscopy (Jeol TEM 1200EX, JEOL Ltd., Tokyo, Japan) at an accelerating voltage of 100 kV. The sample was dispersed on a copper grid with carbon support film.

### 2.6. Antibacterial Activity

The antibacterial activity of HNT/ERY before and after removing residual ERY from HNTs was assayed. The antibacterial activity of PCL/HNT/ERY nanofibers was assessed with the diffusion method by placing the HNT, HNT:ERY, 1 cm^2^ of prepared PCL fiber, PCL/HNT fiber, and PCL/HNT and ERY fiber on the inoculated (10^6^ cells/mL) MHA (Mueller Hinton Agar) growth media. For inoculation, the model bacteria *S. aureus* CCM 3953 (Czech Collection of Microorganisms) and *E. coli* CCM 3988 were used. The antibacterial activity of prepared nanofibers was evaluated as the occurrence of a sterile zone of inhibition around the modified HNTs with ERY and modified fibers of PCL/HNT and ERY, respectively, after cultivation for 24 h at 37 °C. With the aim to wash out the residual ERY from the surface of the prepared materials, modified HNTs with ERY and the PCL/HNT and ERY fibers were washed two times in methanol and two times in water. The antibacterial activity was evaluated again as described above. To evaluate the sustained release of ERY from the PCL/HNT fiber, serial cultivation was examined as follows. The fresh prepared PCL/HNT and ERY fibers were first placed on the inoculated growth media, and the antibacterial effect was evaluated after 24 h of cultivation at 37 °C. Then, these same fibers from the grown bacterial cultures were replaced with fresh ones that were again cultivated and considered. This procedure was repeated for 5 days.

### 2.7. Mechanical Property Measurement

Mechanical properties were tested in uniaxial tension at a crosshead speed of 5 mm·min^−1^ and ambient temperature of 22 °C using a Zwick Roell Z010 (Zwick-Roell, Ulm, Germany) equipped with a 10 kN load cell. Six specimens of rectangle shape (approximately 30 × 5 mm^2^) were tested for each type of material, and the averages and standard deviations were determined. The thickness of the specimens was measured with a micrometer.

### 2.8. Thermogravimetric Analysis

Thermal stability was determined by thermogravimetric analysis (TGA) using a Q1500 D instrument (from MON Budapest, with TA Universal Analysis software). Each sample (100 mg) was heated from 30 to 600 °C at a heating rate of 10 °C/min in the presence of air with a flow of 50 mL/min. The corresponding weight loss was recorded as a function of temperature.

## 3. Results and Discussion

### 3.1. Microscopic HNT/ERY’ Structure Analysis

As our task was the preparation of antimicrobial nanofibers for biomedical applications, the main criterion for the selection of an appropriate type of halloysite was its purity. For this reason, we selected ULTRA HalloPureTM tubular halloysite. This particular HNT type has a very high purity (93.5%), with only trace levels of feldspar. The second reason for our study was ULTRA HalloPure’s geometry. From the point of view reinforcing effect, which is essential for nanofibers based on low-strength PCL, it is necessary to use HNTs with long aspect ratios. Moreover, long-aspect-ratio HNTs provide more prolonged antibacterial effects due to the release of an active compound. The geometry of the used HNT type (length, inner and outer diameter, and aspect ratio) is presented in Table 1.

The micromorphology and nanomorphology of HNTs both modified and loaded with ERY were studied with transmission electron microscopy. The structure of HNTs before ERY loading is shown in Figure 2. TEM images of HNTs loaded with ERY are shown in Figure 3.

### 3.2. Loading Efficiency of HNTs

A thermogravimetric study of HNTs and HNT/ERY was performed to determine loading efficiency, i.e., the amount of ERY loaded into HNTs. As mentioned earlier, the loading of active agents to HNTs is based on the diffusion of molecules from external solutions into the inner part of HNTs due to the concentration gradient. As shown in Figure 3, the unmodified HNTs were stable up to 500 °C, at which point de-hydroxylation [6,39] occurred with a disruption of the tube-wall multilayer packing. At 600 °C, the weight loss of the unmodified HNTs was 12.5 wt.% (blue curve in Figure 4). Pure erythromycin showed steep decomposition around 300 °C followed by a slow degradation process up to 600 °C. (red curve in Figure 4). The efficiency of the used loading method was evaluated via a comparison TGA of pristine HNTs with HNTs loaded with ERY (HNT/ERY) washed with water (green curve in Figure 4) and methanol (orange curve in Figure 4).

The sample of HNT/ERY washed in water (HNT/ERY/WAT) showed the main decomposition around 300 °C followed by gradual degradation process and further degradation step around 500 °C, reflecting the behavior of precursors.

The weight loss of HNT/ERY/WAT—37.9 wt.% at 500 °C—corresponded to the rate of erythromycin loaded on both the inner and outside surface of HNTs (HNT:ERY at 60:40) and the rate of ERY loaded exclusively onto the inner surface of HNTs, as evaluated on the sample from which residual ERY was removed with methanol: 4.6 and 11.5 vol.%, respectively. In summary, we obtained extremely highly-loaded HNTs suitable for antibacterial applications in biomedicine.

### 3.3. Morphological Observation of PCL/HNT/ERY Nanofibers

Our previous study described the morphology of electrospun PCL antibacterial composites with different fillers—hydrophobic quantum dots working on the principle of photodynamic therapy—for wound healing in tissue engineering in detail [29]. A strongly porous material with interconnected structures was obtained via the electrospinning method. This 3D structure is very suitable for use in tissue engineering. Despite this fact, we created relatively homogeneous nanofibers with an average thickness of 3–4 μm, and the same effect was not observed in the case of HNT-loaded samples. As shown in Figure 5, the fibers contained a lot of beads in their structure due to the very high probability of the electric conductivity of the electrospinning solution to change, which is one of the key factors of the electrospinning process. Hydrophobic quantum dots are semiconductors, and HNTs or ERY-loaded HNTs are insulators. Therefore, the final electric conductivity of the solution was significantly lower and the viscosity of solution was reduced. All those parameters affected the final structure of the fibers. These beads had only a minor effect on antibacterial and mechanical properties because these were determined by releasing ERY from HNTs, as shown later.

### 3.4. Antibacterial Properties of PCL/Halloysite/ERY

The antibacterial activity of HNT/ERY before and after the removal of residual ERY from HNT surfaces was assayed (Figure 6), and the obtained antibacterial properties of PCL/HNT and ERY are shown in Figure 7. Because PCL and HNT/PCL were not antibacterial active, as is clear from Figure 7, no inhibition zone was formed; the HNT/ERY and PCL/HNT/ERY nanofibers revealed significant antibacterial activity.

Zones of inhibition were observed around the HNT and ERY fibers and the = PCL/HNT and ERY fibers. After the successful incorporation of ERY into the carrier HNTs (proving that it was also a part of the prepared PCL/HNT fibers), the washing of the fibers in methanol followed by washing in water was conducted to ensure that the whole residual ERY of the surface was removed. Next, the antibacterial assay was repeated with washed materials. As is clearly shown in Figure 6 and Figure 7, even though the zones of inhibition of washed samples were smaller, antibacterial activity was recorded again, which means that the antibiotic was incorporated into the HNT nanostructures. Accordingly, HNTs are an appropriate carrier for ERY.

Finally, the antibacterial activity was assayed when the PCL/HNT and ERY fibers were repeatedly (three times) replaced on fresh inoculated growth media. The results are shown in Table 2. These results support the idea of sustained release from the prepared fiber because the antibacterial effect was still observed after the second replacement (72 h) for *E. coli* and after the first replacement (48 h) for *S. aureus* (Table 2).

### 3.5. Mechanical Properties of PCL/HNT/ERY Nanofibers

As it is clearly evident, the mechanical properties of investigated PCL nanofiber system were significantly influenced by the presence of both unmodified and HNT-loaded ERY (Figure 8, Table 3). The introduction of HNT particles into nanofibers significantly increased the Young’s modulus and tensile strength. Furthermore, elongation at break remained unchanged for this system. The observed enhancement correlated well with the results obtained on PCL [40]. Stress transfer, volume replacement, and segmental immobilization [41,42,43] reinforcing mechanisms were responsible for this enhancement of mechanical performance. The loading of ERY into HNTs decreased Young’s modulus to the value of the PCL while elongation at break was significantly increased. The observed trend re-affirmed the effect of plasticizers—small compatible molecules that are able to decrease the glass transition temperature of polymers—and suggested the good compatibility of ERY with the PCL/HNT nanofibers. More importantly, tensile strength also decreased upon ERY loading. However, the decrease was so slight that the values of the PCL/HNT and ERY systems were still more than 100% higher than those of the pure PCL system. A similar tensile strength enhancement (about 100%) was observed for PCL/gelatin/HNT microfiber system loaded with metronidazole [44].

## 4. Conclusions

In this work, antibacterial, biodegradable, environmentally friendly, and nontoxic nanofibers with good biocompatibility and cost-effective production were prepared via the electrospinning of PCL and HNTs loaded with ERY. The PCL/HNT/ERY nanofibers exhibited outstanding antibacterial properties and resulted in the inhibition of both Gram-negative (*Escherichia coli*) and Gram-positive (*Staphylococcus aureus*) bacteria. We observed the gradual release of ERY out of the nanofibers, that was detected as repetitive antimicrobial activity after the material replacement on fresh prepared inoculated growth media. In addition to the antibacterial properties, the significant enhancement of mechanical properties was achieved via the incorporation of both unmodified and ERY-loaded HNTs, thus improving suitability of the system for medical applications. To summarize, PCL/HNT/ERY nanofibers with strong antibacterial effects have great potential as a progressive new biomedical material in both tissue engineering and a number of other biomedical applications.

## Figures and Tables

**Figure 1 polymers-14-00746-f001:**
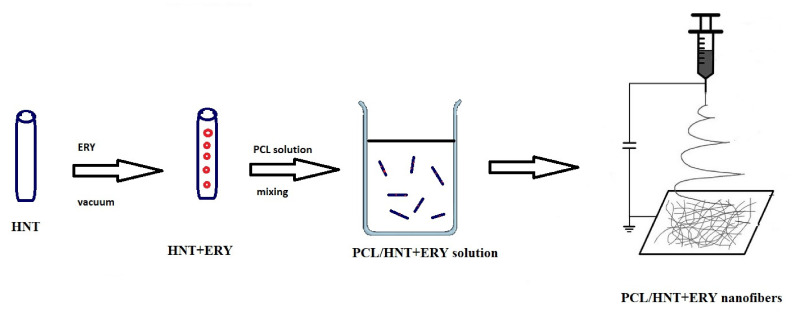
Scheme of electrospun nanofiber preparation.

**Figure 2 polymers-14-00746-f002:**
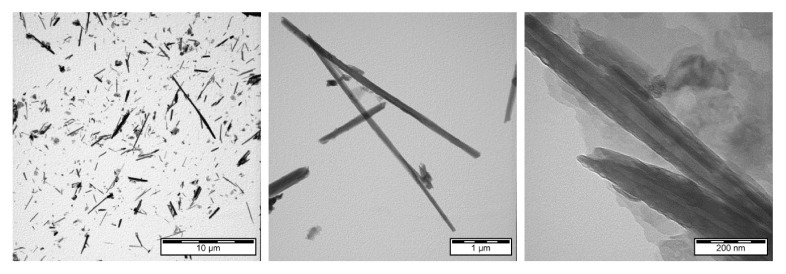
TEM of unmodified HNTs before ERY loading.

**Figure 3 polymers-14-00746-f003:**
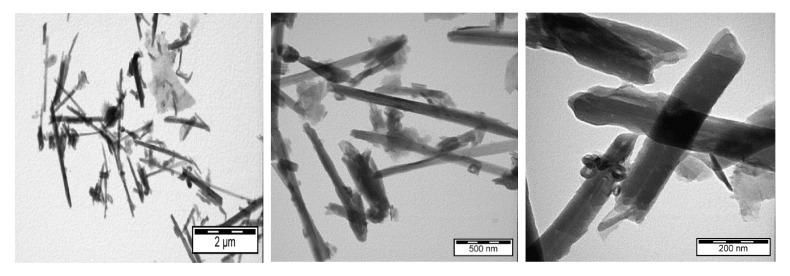
TEM of HNTs loaded with ERY.

**Figure 4 polymers-14-00746-f004:**
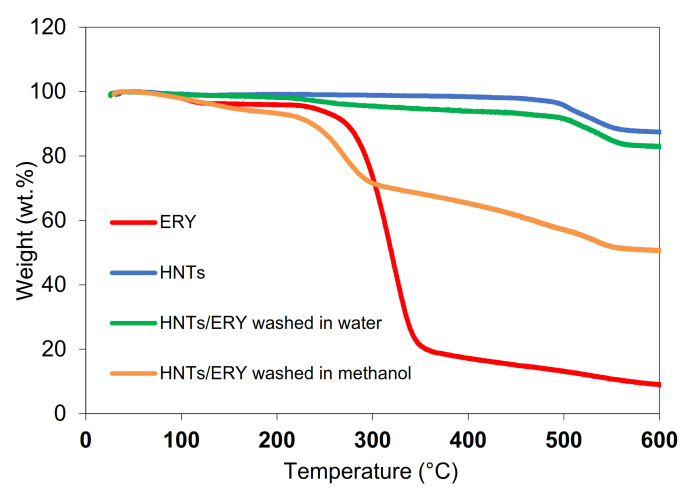
TG curves of ERY, HNTs, HNT/ERY washed in water, and HNT/ERY washed in methanol.

**Figure 5 polymers-14-00746-f005:**
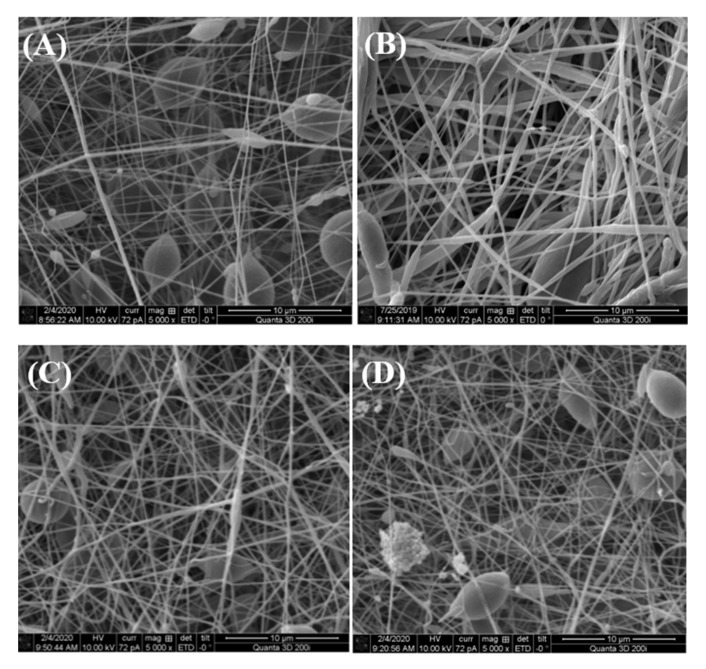
SEM images at high magnification (5000×) in the secondary electron mode of electrospun (**A**) pure PCL, (**B**) PCL and 6 wt.% HNT nanofibers, (**C**) PCL and 6 wt.% HNT/ERY (80:20), and (**D**) PCL and 6 wt.% HNT/ERY 60:40.

**Figure 6 polymers-14-00746-f006:**
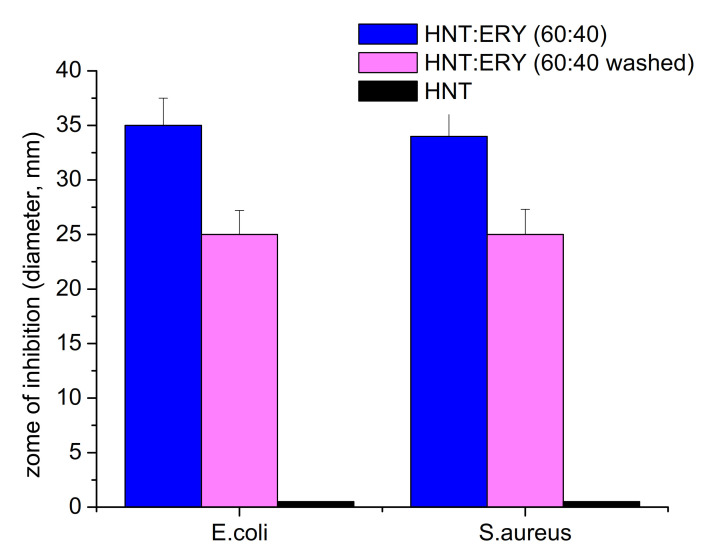
Comparison of the antibacterial activity of HNT/ERY before and after removing ERY from the outside surface of HNTs assayed with the disk diffusion method on *E. coli* and *S. aureus*. The antibacterial activity was recorded as the inhibition zone diameter ± SE (standard error).

**Figure 7 polymers-14-00746-f007:**
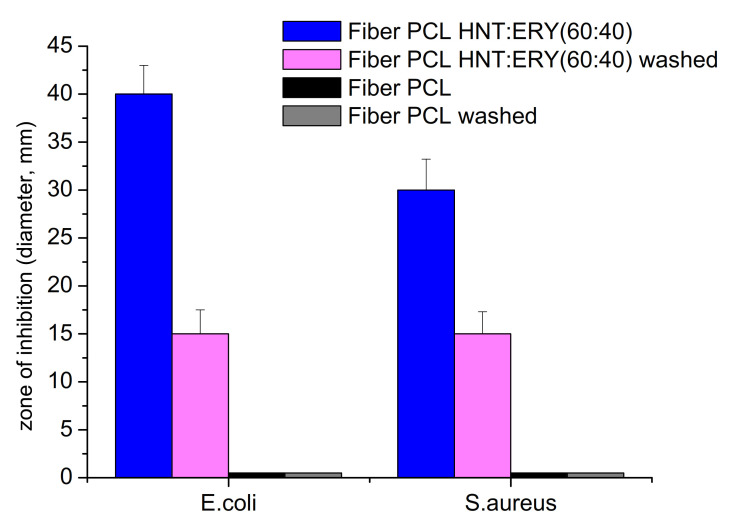
Comparison of the antibacterial activity of washed and unwashed PCL/HNT fibers with ERY assayed with the disk diffusion method on *E. coli* and *S. aureus.* The antibacterial activity was recorded as the inhibition zone diameter ± SE (standard error).

**Figure 8 polymers-14-00746-f008:**
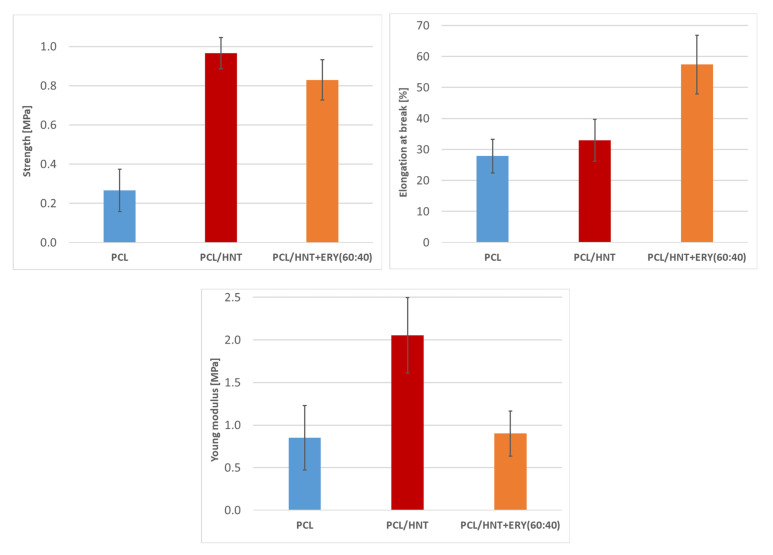
Tensile strength, elongation at break, and Young’s modulus of PCL, PCL/HNT, and PCL/HNT and ERY nanofibers.

**Table 1 polymers-14-00746-t001:** Structural parameters of ULTRA HalloPure.

Length (µm)	Inner Diameter (nm)	Outer Diameter (µm)	Aspect Ratio
1.0–2.0	15–20	0.10–0.20	Typically 15

**Table 2 polymers-14-00746-t002:** Antibacterial activity of PCL/HNT and ERY fiber after 3x replacement.

Model Bacteria	Zone of Inhibition (mm)		
	Mode of Application	Fiber PCL/HNT	Fiber PCL/HNT and ERY
** *E. coli* **	Placed on the growth media	0	40
	Replaced for the 1st time	0	35
	Replaced for the 2nd time	**0**	**21**
	Replaced for the 3rd time	0	0
** *S. aureus* **	Placed on the growth media	0	30
	Replaced for the 1st time	**0**	**30**
	Replaced for the 2nd time	0	0
	Replaced for the 3rd time	0	0

**Table 3 polymers-14-00746-t003:** Young’s modulus, tensile strength, and elongation at break of PCL, PCL/HNT, and PCL/HNT and ERY samples.

Sample	Young’s Modulus (MPa)	Strength (MPa)	Elongation at Break (%)
**PCL**	0.9 ± 0.4	0.3 ± 0.1	27.9 ± 5.4
**PCL/HNT**	2.1 ± 0.4	1.0 ± 0.1	32.9 ± 6.8
**PCL/HNT and ERY(80:20)**	0.7 ± 0.4	0.76 ± 0.1	56.8 ± 9.2
**PCL/HNT and ERY(60:40)**	0.9 ± 0.3	0.83 ± 0.1	57.4 ± 9.5

## Data Availability

The data presented in this study are available on request from the corresponding author.

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
