# Peer review of "Antibacterial Electrospun Polycaprolactone Nanofibers Reinforced by Halloysite Nanotubes for Tissue Engineering"

_polymers, 2022, doi:10.3390/polym14040746_

Round 1
Reviewer 1 Report
Summary
The manuscript entitled “Antibacterial electrospun polycaprolactone nanofibers rein-2 forced by halloysite nanotubes for tissue engineering’’ by Khunova et al. shows the development of antibacterial nanofibers based on PCL and halloysite nanotubes. The presented article contains both basic experimental and application tests, which are vital to prove the claim with reliable data and in turn, pave the way towards the application of this system.
General comments
In general, the work is not very accurate and well presented as Polymers articles should be anyway the presented results are certainly of interest to readers of this journal. The article style is not correct, and it should be reviewed in a few points. Thus, I believe that the text needs some technical adjustments to be published. Therefore, I recommend that this manuscript can be published in Polymers after Major revision.
Specific comments
Going into details on the specific issues, here some comments are reported:
- the article's grammar, punctuation, and style are very poor, and the manuscript needs to be deeply proofread. The structure of the article is odd and sounds more like a report than a scientific article. The last paragraph of the Introduction should be dedicated to describing the presented research, but this part is not present.
- the use of inorganic nanotubes as electrospun nanofiber fillers can find applications out of the biomedical field too (e.g., [Fibers and Polymers , Volume 16, pp 426-433, 2015 - https://doi.org/10.1007/s12221-015-0426-x]). Please mention it in the Introduction
- all the quantitative result should be reported, including statistical information like STD, error bars etc etc. (e.g., there is no statistics connected to antibacterial test)
- Figure 3. First of all, there is a typo in the caption since DTA is not mentioned. Moreover, it is unclear why DTA is strongly rising before 400C if the TGA does not show any consistent weight variation.
- I do not understand why the final application test (antibacterial analysis) as not reported at the very end of the manuscript.
- This research work suffers from a fundamental problem. When electrospinning research starts, the first step is material optimization to avoid structural defects. Unfortunately, the authors skipped this step and moved to the next step. Indeed, the presented mats are completely affected by an enormously high number of beads. I`m very sorry to say that the presented data are not publishable. The authors need to prepare the entire set of materials (of course without beads or any other defects) again and perform all the shown tests again
Conclusion
The topic of this manuscript falls within the scope of Polymers. I like the concept and material development/characterization proposed in this paper. Anyway, I was very torn during the preparation of this report. The presented data are extremely affected by the defects present in the fabricated mats, which make the presented research activity not good enough to deserve publication. On the other hand, the idea is interesting, and I believe the authors merit a second chance. Finally, I believe the article is of sufficient novelty to meet the Polymers publication standard after (a very) major revision.
Reviewer 2 Report
The work submitted by Viera Khunova et al. reported an antibacterial fibers prepared by electrospinning. The as-prepared fibers with HNTs/ERY show certain antibacterial properties. I would suggest the work published in polymers after some revisions.
- The work only evaluated the antibacterial activity of PCL nanofibers with 7.0 wt.% HNT/ERY content. I suggest adding the experiments of antibacterial activity of PCL nanofibers with different HNT/ERY contents nanofibers.
- The authors state that the prepared fibers is a non-toxic material, but we know that HNTs nanotubes may be harmful to human body if they fall off from the fiber more than a certain amount. There is no evaluation of the toxicity of the material in the manuscript.
- P2L75, the description of electrospinning is an old technology is wrong.
- It is recommended to draw a diagram of the antibacterial fibers preparation process.
- P3L145, the formatting of the title is wrong. P4L169, change mmmin-1 to mm·min-1. P6L212, the title of the Fig. 3 is wrong.
- P5L206, “The efficiency of the used loading method was evaluated by comparison TGA of pristine HNTs (orange curve) with HNT/ERY (blue curve) washed with water and methanol.” Please check this sentence carefully.
- The authors did not analyze the TEM images before and after ERY loading.
- P6L238, the final structure of fibers may also be affected by the change of solution viscosity.
- The format of reference 8 is incorrect. Please carefully check the format of all references.
Round 2
Reviewer 1 Report
-
Reviewer 2 Report
I suggest that the manuscript can be accepted and published in polymers.